# Governance in Crisis: A Mixed-Methods Analysis of Global Health Governance During COVID-19

**DOI:** 10.3390/ijerph22081305

**Published:** 2025-08-20

**Authors:** Kadria Ali Abdel-Motaal, Sungsoo Chun

**Affiliations:** Institute of Global Health and Human Ecology, American University, New Cairo 11835, Egypt; kkmotaal@aucegypt.edu

**Keywords:** global health governance, pandemic response, COVID-19, governance effectiveness

## Abstract

Background: The COVID-19 pandemic exposed major structural deficiencies in global health governance, including stark inequities in vaccine access, intervention timing, and mortality outcomes. While economic resources played a role, the influence of governance performance remains insufficiently examined. This study addresses a significant gap by integrating governance metrics with pandemic response data to assess how governance quality, independent of income level, affected national outcomes. Although the Oxford COVID-19 Government Response Tracker (OxCGRT) dataset has been widely used to document policy responses, this study offers a novel contribution by linking these policy interventions with governance performance and evaluating their joint effect on health outcomes and vaccine equity. Methods: This mixed-methods study combines quantitative analysis of global datasets with a qualitative literature review. Quantitative data were mainly obtained from the Oxford COVID-19 Government Response Tracker (OxCGRT), the World Bank’s Worldwide Governance Indicators (WGIs), and World Bank/WHO databases. A governance performance index was constructed using two WGI components: Government Effectiveness and Regulatory Quality. Countries were grouped into high, medium, or low governance categories. Statistical tests included ANOVA, Kaplan Meier survival analysis, and multivariable OLS regression. The qualitative component reviewed 45 academic and institutional sources on governance performance during COVID-19. Results: Countries with high governance performance had earlier public health interventions, lower mortality, and broader vaccine coverage, independent of income level. Kaplan Meier analysis revealed faster school closures in these countries (*p* < 0.01). Multivariable regression showed governance remained a significant predictor after adjusting for income and health spending. Qualitative findings highlighted recurring weaknesses in legal enforceability, intergovernmental coordination, and global financing mechanisms. Conclusions: Governance performance had a decisive impact on pandemic outcomes. The COVID-19 crisis revealed the need for robust governance systems capable of responding to complex emergencies that extend beyond the health sector into institutional, economic, and social spheres.

## 1. Introduction

### 1.1. Background and Rationale

Global health governance (GHG) refers to the collective efforts, institutions, rules, and processes through which international and transnational actors coordinate actions to address global health challenges [1]. It involves formal structures such as treaties, international health regulations, and organizational mandates, as well as informal collaborations among states, international organizations, non-governmental organizations (NGOs), and private sector entities. Effective GHG is essential for managing health risks that transcend national borders and require coordinated global responses.

Kickbusch [2] highlights that GHG reflects the interconnected nature of health determinants influenced by globalization, such as international trade, travel, climate change, and migration. She identifies three core features of GHG: (1) the global scope of health threats, (2) the role of transboundary factors influencing health outcomes, and (3) the need for multisectoral and multilevel collaboration involving both public and private actors.

The COVID-19 pandemic exposed significant gaps and inefficiencies in global health governance systems. These included unequal access to vaccines, uncoordinated responses between countries, fragmented funding mechanisms, and limited transparency and accountability in decision-making processes [3]. Although global institutions such as the World Health Organization (WHO), the Vaccine Alliance (Gavi), the COVID-19 Vaccines Global Access (COVAX), and the International Monetary Fund (IMF) played central roles in pandemic management, their efforts were often constrained by political tensions, resource limitations, and structural weaknesses [4]. These challenges have raised urgent questions about the adequacy of current GHG frameworks in responding to global health emergencies.

This study employs a mixed-methods approach to evaluate the performance of global health governance during the COVID-19 pandemic, with particular focus on its strengths, weaknesses, and limitations. The quantitative analysis examines disparities in pandemic outcomes, such as vaccine distribution and mortality rates, across regions and income levels. The qualitative component explores institutional challenges, governance breakdowns, and policy gaps through document analysis and expert insights. By integrating both data-driven and interpretive analyses, this research aims to provide a comprehensive evaluation of GHG performance and contribute evidence-based recommendations for improving future pandemic preparedness and response.

### 1.2. Research Problem

The COVID-19 pandemic revealed persistent global disparities in vaccine distribution, access to essential medical supplies, and mortality outcomes. These challenges have raised important concerns about the effectiveness of global health governance (GHG) structures, particularly regarding transparency, coordination, and accountability.

Although prior research has examined health system preparedness and institutional failures during the pandemic, there is limited empirical analysis assessing how governance quality influenced key pandemic outcomes. Furthermore, the differential impact of governance structures across centralized and decentralized systems remains underexplored. This study addresses a significant gap by integrating governance metrics with pandemic response data to assess how governance quality, independent of income level, affected national outcomes.

Although the Oxford COVID-19 Government Response Tracker (OxCGRT) dataset has been widely used to document policy responses, this study offers a novel contribution by linking these policy interventions with governance performance and evaluating their joint effect on health outcomes and vaccine equity.

### 1.3. Primary Research Question

How did the global health governance system influence the COVID-19 response, particularly in relation to vaccine distribution, coordination of interventions, funding mechanisms, and mortality outcomes?

#### Sub-Questions

What limitations in global health governance affected decision-making, mortality rates, and equitable vaccine distribution during the pandemic?How did governance effectiveness, independent of income classification, influence pandemic response outcomes across centralized and decentralized systems?What policy reforms are needed to strengthen global health governance for future pandemics?

## 2. Materials and Methods

This study adopts a mixed-methods design to assess the effectiveness and limitations of global health governance (GHG) during the COVID-19 pandemic. Quantitative analysis of cross-national data, covering policy response timing, mortality rates, and vaccine distribution, is integrated with qualitative synthesis of scholarly literature, policy documents, and institutional reports. This triangulated approach enhances analytical validity and enables both empirical and contextual examination of governance dynamics.

A coding framework was developed to guide the qualitative strand, identifying recurring themes such as vaccine equity, funding mechanisms, and coordination efficiency. Documents were reviewed in iterative coding cycles to ensure thematic saturation and reliability. In parallel, statistical methods (e.g., ANOVA, regression, Kaplan–Meier survival analysis) were employed to identify governance-related patterns in health outcomes.

Findings from both strands were synthesized using a structured integration framework to facilitate cross-validation and deeper interpretation of governance impacts.

The sections that follow present the qualitative and quantitative components of this study, each contributing to a comprehensive understanding of governance effectiveness during the pandemic.

### 2.1. Qualitative Component

This section presents the qualitative component of this study, which explores governance-related factors through thematic analysis of selected literature and institutional sources.

#### 2.1.1. Study Design

This component is based on a purposive review approach, where literature is selected based on its direct relevance to the research question. A purposive review is a targeted method that prioritizes depth over breadth, allowing for a focused synthesis of key themes, governance mechanisms, and policy frameworks [5]. Unlike systematic reviews, which aim for exhaustive coverage, purposive reviews focus on the most informative sources to provide meaningful insights [6]. The goal is to examine the effectiveness of global health governance in pandemic response and its limitations, with a focus on vaccine disparities, response coordination, funding distribution, and governance models. The review process is guided by a structured search strategy, clearly defined inclusion and exclusion criteria, and a thematic coding framework, as detailed in the following subsections.

#### 2.1.2. Search Strategy and Selection Criteria

To ensure a rigorous and comprehensive review, a structured search was conducted across major academic and policy databases, including PubMed, Scopus, Web of Science, and Google Scholar, as well as institutional reports from the WHO, the World Bank, the IMF, and the Oxford COVID-19 Government Response Tracker. The following Boolean search terms were used: “Global health governance” AND “COVID-19”; “Pandemic response” AND “governance effectiveness”; “Vaccine equity” OR “vaccine disparities”; “International health regulations” AND “pandemic preparedness”. To ensure a broad yet focused review, truncation was applied (e.g., govern* to include governance, governing, and government).

The time frame was a five-year lookback (2020–2024); it was chosen because governance structures in response to global crises evolve rapidly. As Shulman [7] suggests, policy frameworks undergo significant transformation over time, and a shorter timeframe ensures relevance to current governance challenges.

The inclusion criteria were as follows: articles (1) addressing global health governance mechanisms during the COVID-19 pandemic, vaccine distribution, funding disparities, response coordination, or governance models; (2) published in peer-reviewed journals, policy reports, or reputable institutional publications; and (3) written in the English language.

The exclusion criteria were studies unrelated to global health governance and non-English studies unless translated.

The initial search identified 120 records. After screening titles and abstracts and removing duplicates and non-relevant studies, 60 full-text articles were reviewed. Fifteen were excluded due to lack of relevance to governance structures or limited methodological rigor. The final sample included 45 studies, which were analyzed using a structured coding framework to extract core themes, policy recommendations, and governance mechanisms, (Appendix A Prisma Flow Diagram).

#### 2.1.3. Data Analysis

To ensure alignment between the qualitative and quantitative components, the coding framework was adapted to reflect the core quantitative variables: vaccine equity, response speed, and mortality outcomes across governance performance levels. Although funding distribution was initially examined quantitatively, its results were statistically insignificant and thus excluded from the final statistical analysis. Nevertheless, the theme was retained in the qualitative synthesis due to its conceptual importance. The final coding framework systematically categorized qualitative data under four themes: vaccine equity, response speed, mortality outcomes, and funding distribution, as presented in Table 1.

### 2.2. Quantitative Component

The COVID-19 pandemic was one of the most transformative public health crises since the early 20th century. Governments responded under conditions of acute uncertainty and systemic strain, adopting a range of containment and mitigation strategies with varying speed and intensity, often revised as circumstances evolved. These differences reflected underlying variations in governance structures, administrative capacities, and policy priorities, providing a basis to evaluate the effectiveness of national-level intervention models.

This study employed a cross-sectional quantitative design using secondary data from publicly available global governance and health databases. The analysis aimed to assess the association between governance performance and three key pandemic outcomes:-Policy response timing, as an indicator of governance efficiency.-COVID-19 mortality rates, as a measure of governance effectiveness.-Vaccine coverage disparities, as a proxy for governance equity.

#### 2.2.1. Data Sources and Integration

This study uses the World Bank’s Worldwide Governance Indicators (WGIs), a cross-country dataset that measures governance across more than 200 countries. The indicators are compiled from over 30 data sources, including surveys and expert assessments.

WGI covers six key dimensions, Government Effectiveness, Regulatory Quality, Voice and Accountability, Political Stability, Rule of Law, and Control of Corruption, scored on a scale from 0 to 100, with higher scores indicating better governance performance [8].

In our analysis, we focus on two indicators:-Government Effectiveness (GE): how well a government delivers public services and manages operations.-Regulatory Quality (RQ): how effectively a government creates and enforces policies that support private sector development.

These technocratic indicators emphasize institutional and operational capacity rather than democratic accountability, making them well-suited for assessing public health governance [9,10].

We calculate a composite governance score by averaging each country’s GE and RQ scores, classifying countries into high, medium, and low governance performance levels for comparison. To assess national COVID-19 policy responses, this study used data from the Oxford COVID-19 Government Response Tracker (OxCGRT), which systematically records government interventions and case counts over time across 180 countries [11]. OxCGRT includes 24 standardized indicators across four thematic policy domains: (1) Containment and Closure, (2) Health System Policies, (3) Economic Policies, and (4) Vaccination Policies.

From this dataset, we extract five binary-coded indicators that signal the initiation of major COVID-19 interventions. These triggers reflect the timing of nationally implemented policies and include the following:Facial Coverings (H6M), reflecting mandates or recommendations on mask usage in public.Public Information Campaigns (H1), indicating whether governments launched coordinated COVID-19 awareness efforts.International Travel Controls (C8EV), measuring entry and exit restrictions.*Stay-at-Home Requirements (C6M)*, assessing the extent of government-imposed home confinement measures.School Closures (C1M), capturing the timing of nationwide school shutdowns.

These variables were selected for their relevance to early pandemic containment and comparability across governance contexts.

To ensure data harmonization, we synchronized policy intervention dates relative to each country’s first confirmed COVID-19 case. This enabled standardized comparisons of response timing across countries with varying outbreak timelines.

Key OxCGRT indicators were transformed into binary time-to-event variables, coded “1” on the date of national activation and “0” prior.

The dataset was integrated with the World Governance Indicators (WGIs) using consistent country-level identifiers, with temporal alignment achieved by using 2019 governance scores as a pre-pandemic baseline. Policy and health outcome data were drawn from the 2020 to 2021 pandemic years.

Ordinary Least Squares (OLSs) assumptions were tested using Breusch–Pagan and Durbin–Watson diagnostics, confirming no evidence of heteroskedasticity or autocorrelation.

While variable standardization (e.g., z-scores) was not applied, results were based on comparable scales and robust model performance.

#### 2.2.2. Statistical Analysis

To guide the quantitative analysis, we employed a multi-method statistical approach tailored to each research objective. Table 2 summarizes the key statistical techniques used to evaluate the relationship between governance performance and three pandemic-related outcomes: policy response timing, COVID-19 mortality, and vaccine equity. For each outcome, we specify the variables, statistical tests, and data sources. This approach ensures alignment between research questions, data types, and analytical tools.

Findings from both strands were integrated, allowing the qualitative insights on governance challenges to be substantiated by empirical data, thereby enhancing the validity and contextual depth of the analysis.

## 3. Results

### 3.1. Qualitative

#### 3.1.1. Vaccine Allocation Disparities

##### Decision-Making Processes at Global, Regional, and National Levels

Vaccine allocation strategies varied across global, regional, and national levels, shaped by political, economic, and public health factors. Islam et al. [12] found the US vaccine allocation strategy suboptimal, suggesting that prioritization of individuals with comorbidities would have reduced mortality. Wouters et al. [13] noted LMIC dependence on HICs due to production constraints and IP barriers, advocating for TRIPS reform. Wealthy nations dominated early vaccine access, while LMICs faced delays and, even when vaccines were available, struggled with distribution due to cold chain and workforce limitations.

##### Role of COVAX, WHO, and Bilateral Agreements in Vaccine Distribution

Eccleston and Upton [14] and Puyvallée and Storeng [15] noted that although COVAX aimed to promote global vaccine equity, its effectiveness was undermined by geopolitical interests and vaccine nationalism. High-income countries prioritized bilateral deals, limiting COVAX’s access to supply. Donations were often guided by diplomatic motives rather than equity. COVAX’s reliance on voluntary contributions led to funding gaps and delays. As a result, regional blocs like the African Union pursued independent procurement efforts, exposing structural weaknesses in COVAX’s ability to deliver timely and equitable vaccine distribution.

##### Political and Economic Challenges in Equitable Access

Vaccine nationalism led to stockpiling in high-income countries, delaying equitable access [16]. Pan et al. [17] broadened the concept of vaccine nationalism to include national security, technological competition, and geopolitical interests. Neocolonial economic policies enabled wealthier nations to monopolize supplies, relegating low-income countries to donor-driven mechanisms like COVAX, which often reinforced rather than resolved structural inequities [18]. Distribution disparities stemmed from macro-level factors, economic power, health system capacity, and legal barriers, and micro-level factors such as socioeconomic status, race, gender, and insurance coverage [19].

##### Transparency in Procurement and Allocation

Pharmaceutical pricing mechanisms lacked transparency due to confidential agreements, undisclosed profit margins, and hidden rebates. Although the 1989 EU Transparency Directive required price list publication, it did not mandate disclosure of actual prices after discounts. This information asymmetry weakened LMICs’ negotiating power. Limited access to R&D costs, trial data, and profits obscured vaccine production costs. Empirical evidence on the effectiveness of price transparency policies remains scarce, with national initiatives showing mixed results [20].

#### 3.1.2. Pandemic Response Time

##### Effectiveness of Global Coordination Mechanisms

Elekyabi [21] highlights major limitations in WHO’s capacity to coordinate the global COVID-19 response, particularly its emergency declarations and reliance on voluntary cooperation. While the Public Health Emergency of International Concern (PHEIC) declaration helped raise awareness, its non-binding nature limited WHO’s authority to enforce timely action. Without legal mechanisms to ensure compliance, countries delayed or withheld critical data, hampering global coordination. Political pressures further led many governments to prioritize economic and domestic interests over WHO guidance, especially in the pandemic’s early stages. As a result, national strategies often diverged from international recommendations, undermining cohesive action. Elekyabi calls for strengthening WHO’s legal authority, enhancing transparency, and building national preparedness capacities to improve global coordination in future health emergencies.

##### Factors Causing Delays in International Pandemic Response

The COVID-19 pandemic demonstrated that public health emergencies are not solely biomedical but deeply political, social, and infrastructural. Cortés, Pacheco, and Fronteira [22] argue that Italy’s experience revealed how neglecting social inequalities, institutional fragmentation, and public trust can worsen outcomes despite resource availability. They advocate for equity-based strategies integrating the “One Health” approach. Similarly, Baum et al. [23] found that political coordination was critical: centralized governments managed more coherent responses, while fragmented systems exacerbated inefficiencies and inequities. Agyapon-Ntra and McSharry [24] showed that mask mandates were the most cost-effective NPI, but success varied by context; countries with higher literacy and income support saw greater compliance. Their findings support context-specific interventions and stronger safety nets. Joshi et al. [25] highlighted the promise of blockchain-enabled humanitarian networks for improving resource distribution, though digital divides and trust barriers remain. Collectively, these studies emphasize that pandemic response depends on inclusive governance, socioeconomic resilience, adaptive policy design, and the equitable deployment of technology.

##### Comparison of Centralized Versus Decentralized Governance Models

The governance structure of a country influenced the effectiveness of its pandemic response. Baum et al. [23] emphasize that centralized political systems with strong institutional collaboration were better positioned to implement timely and coordinated actions, whereas decentralized systems often faced fragmented communication, leading to non-compliance and mistrust. Spain offers a nuanced case: initially adopting a centralized approach, it later transitioned to a multilevel governance model. Navarro et al. [26] highlight that political polarization and uneven regional capacities constrained the response. China’s centralized yet multilayered system enabled rapid containment using big data and targeted lockdowns [27]. However, early delays in local reporting (e.g., Wuhan) and limited autonomy revealed structural weaknesses. Ahmad [27] contrasts this with Germany, South Korea, and Australia, where strong health systems and effective coordination led to successful outcomes, unlike in the United States and the United Kingdom, were delayed responses and fragmented systems undermined efficiency. Sun et al. [28] advocate for a globally integrated approach to pandemic governance, noting that despite China’s proactive domestic response, global gaps in data sharing, coordination, and vaccine equity limited collective resilience. In India, GIGA Focus Asia [29] found that centralization neglected regional disparities, while in Italy, Angelici et al. [30] observed that Veneto’s decentralized response outperformed Lombardy’s centralized approach.

#### 3.1.3. Global Health Funding Distribution

##### Governance Mechanisms Behind Funding Allocation

A recurring theme in the literature is the growing fragmentation and complexity of global health financing governance. Traditional state-centric models have shifted toward a pluralistic landscape involving non-state actors, philanthropic foundations, and public-private partnerships. Sridhar and Batniji [31] and Fergus [32] note that discretionary, earmarked funding now dominates over core, flexible allocations. Institutions like the Global Fund and Gavi employ results-based financing, prioritizing donor-driven, disease-specific targets over holistic national needs. The Global Fund’s Country Coordinating Mechanisms (CCMs) aim to ensure inclusive governance by involving civil society, governments, and affected communities [33,34] but often face challenges of power asymmetry and symbolic participation, particularly in authoritarian or resource-constrained settings. The World Bank plays a dual role as both financier and policy influencer. Tichenor et al. [35] and Ruger [36] emphasize that, despite a shift toward health system strengthening, the Bank’s funding and conditionalities continue to reflect donor-country agendas, complicating local ownership and governance coherence.

##### Disparities in Funding Access Between Low- and High-Income Countries

Several studies highlight persistent inequities in global health financing. Sridhar and Batniji [31] found that diseases like HIV/AIDS received disproportionate funding, while child health, non-communicable diseases (NCDs), and health systems remained underfunded. The Gates Foundation prioritized biomedical research, often based in high-income countries, while the World Bank and US government focused more on service delivery in low-income settings. Fergus [32] noted that funding often favors countries strategically aligned with donor interests, reinforcing access disparities. Vertical programs intensify this by prioritizing disease-specific interventions over system-wide support. Moosa [37] and Sobhani [38] linked IMF-imposed austerity and privatization to reduced public health investment in poorer nations. During COVID-19, De Foo et al. [39] observed that high-income countries rapidly mobilized emergency funds, while low-income countries faced delayed disbursements, external aid dependency, and limited absorptive capacity, further widening the equity gap.

##### Role of the IMF, World Bank, and Governments in Financing Global Health Responses

The IMF’s role in pandemic financing remains contentious. Moosa [37] and Sobhani [38] argue that its conditionalities, emphasizing fiscal discipline, wage ceilings, and inflation control, limit public health investments and restrict access to essential services in low-income countries. In contrast, the World Bank has increased its health financing profile, shifting from infrastructure loans to health system strengthening and results-based financing [35,36]. However, concerns persist over donor influence, weak alignment with health ministries, and gaps in inclusive governance. Recipient governments often remain constrained by donor mandates. Although the Global Fund and Gavi promote local ownership, donor dominance in decision-making undermines autonomy. As Goosby [33] notes, financial oversight mechanisms cannot fully replace sovereign control over national health planning and budgeting.

##### Accountability and Transparency in Pandemic-Related Financial Disbursements

Transparency remains a critical gap in global health financing. Sridhar and Batniji [31] call for a standardized, publicly accessible database to track donor disbursements. Fergus [32] highlights the opacity of financial flows, often obscured by non-transparent funding channels and informal agreements that evade standardized oversight. The Global Fund stands out for its relative transparency and accountability, with audit systems, performance-based funding, and participatory review processes, as described by Goosby [33] and Kavanagh and Chen [34]. Still, limitations persist, particularly in harmonizing reporting requirements with national systems, leading to duplication and inefficiency.

The COVID-19 pandemic brought renewed attention to accountability. De Foo et al., [39] argue that countries with strong governance mechanisms, digital tools, and community engagement managed pandemic financing more effectively. However, many others faced fragmented responses due to delayed donor support, lack of trust, and poor interagency coordination.

#### 3.1.4. Mortality Rates by Governance Model

##### Impact of Public Health Interventions on Mortality Rates

Pandemic outcomes varied significantly across countries with similar containment measures, largely due to differences in governance quality, responsiveness, and cultural alignment. Lim, et al., [40] found that Japan, New Zealand, Vietnam, and Thailand achieved notably lower COVID-19 mortality rates than Italy and Spain, despite applying comparable interventions like lockdowns and travel restrictions. The key differences lie in policy agility, institutional trust, and public compliance. High-performing countries combined rapid, flexible action with transparent communication, decentralized decision-making, and community engagement. In contrast, delayed responses, politicized messaging, and top-down approaches lacking local involvement were associated with poorer outcomes.

##### Governance Model’s Influence on Pandemic Response

Susumu [41] argues that what matters in crisis response is not that the regime is democratic but how well its institutions function. Democracies with high institutional trust and adaptive, transparent governance, such as South Korea and Germany, outperformed autocracies and fragile democracies. Core strengths included civil liberties, strong public institutions, and trusted leadership, which fostered cooperation and policy legitimacy. Nogales Crespo et al. [42] compare Portugal and Spain to show how governance structure influenced mortality. Portugal’s centralized approach enabled early emergency declarations and consistent national policy, resulting in lower death rates. In contrast, Spain’s decentralized system, characterized by political friction and strong regional autonomy, delayed lockdowns and led to fragmented responses. These governance failures were especially evident in Madrid, correlating with higher mortality rates.

A consistent theme across these studies is that responsiveness, coherence, and public trust are critical to reducing pandemic mortality. Effective governance goes beyond policy design; it depends on timely implementation, transparent communication, and whether the public perceives interventions as legitimate and worth following.

Table 3 provides an overview of selected studies that illustrate key dimensions of global pandemic governance. These studies are grouped into four thematic areas: vaccine allocation disparities, pandemic response time, global health funding distribution, and mortality rates by governance model. Together, they highlight the diverse methodological approaches and policy debates that inform current discussions on equity, preparedness, and institutional reform.

### 3.2. Quantitative Results

The quantitative results are organized around three core areas of analysis: (1) the relationship between governance performance and the timing of national policy responses, (2) governance-linked variation in COVID-19 mortality outcomes, and (3) disparities in vaccine distribution as a proxy for governance equity. Each subsection presents findings corresponding to these dimensions.

#### 3.2.1. Governance Performance and Response Timing

Building on the harmonized dataset described earlier, this section examines how national governance performance influenced the timing of key public health interventions during the early months of the COVID-19 pandemic. Using Kaplan–Meier survival curves and ANOVA tests, we assess whether countries with higher governance scores adopted containment measures, such as school closures, stay-at-home orders, and public information campaigns, more rapidly than those with lower governance capacity. Figure 1, Figure 2, Figure 3 and Figure 4 provide comparative timelines of policy implementation across selected country pairs, highlighting variations in response strategies under different governance models and levels of institutional capacity.

In Figure 1 we compare the average implementation timing of five major public health policies: Facial coverings, public information campaigns, international travel controls, stay-at-home requirements, and school closures across countries grouped by governance performance: High, medium, and low. The horizontal axis shows the mean date each policy was first adopted (from January to June 2020), and the vertical axis lists the policy types. Each colored dot represents the average timing for one governance category (performance level). The visual highlights a general trend: countries with stronger governance responded more quickly with public information campaigns and non-pharmaceutical interventions. However, they were slower to impose international travel restrictions, potentially due to greater confidence in domestic containment measures.

In Figure 2, we compare the timing of the first major COVID-19 policy interventions between the United States and China. The policies represented include school closures, stay-at-home requirements, international travel controls, public information campaigns, and facial covering mandates.

The figure shows that China implemented all five policies earlier than the United States, with most measures introduced between January and early February 2020. In contrast, the United States began implementing comparable interventions several weeks later, primarily in late February and March.

This timeline highlights the contrast in early-stage response strategies between the two countries. China’s earlier adoption of public health measures reflects a centralized governance model with swift decision-making, while the United States experienced delays likely influenced by a decentralized federal system, with varied response timing across states and agencies [9].

In Figure 3, we compare the timing of the key COVID-19 policy interventions between Norway and Sweden across five categories: facial coverings, public information campaigns, international travel controls, stay-at-home requirements, and school closures.

The data reveals significant differences in the timing and sequencing of measures. Norway implemented most policies early, particularly school closures and stay-at-home requirements in March 2020, aligning with a more precautionary public health approach. Sweden, in contrast, delayed the implementation of several key interventions, most notably facial covering recommendations, which were not introduced until early 2021.

These differences reflect the contrasting pandemic strategies of the two countries. Norway adopted a more interventionist approach, favoring early restrictions, while Sweden pursued a less restrictive strategy, emphasizing voluntary compliance and maintaining open schools longer. The timing gaps in policy adoption underscore how governance choices influenced the pace and nature of pandemic response even among neighboring countries with similar institutional capacities [8].

In Figure 4, we compare the timing of initial COVID-19 policy interventions in Egypt and Germany, covering five key measures: school closures, stay-at-home requirements, international travel controls, public information campaigns, and facial covering mandates.

The chart shows that Germany generally implemented these interventions earlier than Egypt, particularly in terms of public information campaigns and school closures.

This timing discrepancy suggests differences in institutional agility, public health strategy, and possibly governance structure. Germany’s earlier interventions may reflect its stronger institutional coordination and health system integration, while Egypt’s timeline may point to delayed risk perception or logistical constraints in rolling out nationwide mandates.

A one-way ANOVA was conducted to test whether intervention timing differed significantly between two governance performance levels (high vs. low). The governance variable had 1 degree of freedom (2 groups − 1), and the residual degrees of freedom were 139 (*n* = 140 − 1), allowing for variance partitioning into between-group and within-group components. Table 4 presents the results. With a *p*-value of 0.01, the difference is statistically significant at the 5% level, supporting the visual trends observed in Figure 1, Figure 2, Figure 3 and Figure 4 and confirming that governance performance is meaningfully associated with the speed of policy response.

To further assess the association between governance performance level and the timing of public health interventions, we employ Kaplan–Meier survival analysis. This non-parametric method, unlike ANOVA, does not assume normality and is ideal for time-to-event data. In this context, “survival” denotes the time a country remains without implementing a given intervention, such as school closures. The Kaplan–Meier estimator calculates the probability of survival beyond time t using the formula:

At each time point *t_i_* at which an event occurs:$$S_{t}=\prod_{t_{i}\leq t}(1+\frac{d_{i}}{n_{i}})$$
where

*S_t_* is the probability that a country has not yet implemented the intervention by time *t*.*d_i_* is the number of countries that adopted the intervention at time *t_i_*.*n_i_* is the number of countries still at risk (i.e., those yet to implement the intervention) just before a specific point in time, *t_i_*.

The resulting survival curve illustrates the cumulative probability that countries have not yet enacted school closures over time. A downward step in the curve indicates that some countries have initiated school closures at that point, reducing the proportion still “surviving” without the policy. This approach allows for both visual and statistical comparison of intervention timing across governance performance levels, offering insight into how institutional capacity influences pandemic responsiveness.

We selected school closures as the reference intervention for Kaplan Meier survival analysis, as it was among the earliest and most consequential non-pharmaceutical measures deployed during the COVID-19 pandemic. School closures aimed to reduce viral transmission by limiting large indoor gatherings and close contact among children [43].

Figure 5 presents Kaplan Meier survival curves stratified by governance performance levels (high, medium, and low). The Y-axis represents the probability that school closures had not yet been implemented (survival probability), while the X-axis indicates the number of days since the first confirmed case in each country.

The curves illustrate how quickly countries within each governance tier implemented school closures. The divergence is most pronounced between the high and low governance groups, with high-performing countries implementing closures significantly earlier. Differences between the high and medium tiers are more modest. The accompanying risk table summarizes the number of countries still at risk (i.e., those that had not implemented school closures) at selected time points. For instance, among high-governance countries, 42 were at risk on day 0, with 18 remaining by day 10. In contrast, 55 low-governance countries began at risk on day 0, with 32 still yet to implement school closures by day 10.

The observed variation in school closure timing across governance performance levels is statistically significant, as evidenced by a log-rank test *p*-value of 0.007. This allows rejection of the null hypothesis of no difference at the 5% significance level. To test robustness, we repeated the analysis using alternative combinations of governance indicators from the Worldwide Governance Indicators (WGIs). Results remained consistent, suggesting that the findings are not sensitive to specific governance constructs. This stability likely reflects the strong positive correlations among WGI dimensions, where countries scoring highly in one area (e.g., government effectiveness) also tend to score highly in others (e.g., regulatory quality and rule of law). Overall, the findings validate the hypothesis that stronger governance performance is associated with timelier implementation of public health interventions, with the clearest differences observed between high- and low-performing governance groups.

#### 3.2.2. Governance Performance and Health Outcomes (Mortality)

Building on earlier findings linking governance to the timing and quality of interventions, we now assess its association with COVID-19 mortality, a definitive indicator of pandemic impact. A cross-national dataset was constructed by merging the Oxford COVID-19 Government Response Tracker (OxCGRT), Worldwide Governance Indicators (WGI), and macroeconomic data from the World Bank. These datasets were harmonized at the country level to explore the relationship between governance performance, structural conditions, and reported COVID-19 mortality rates (deaths per 100,000 population).

Some notable anomalies emerged among large developing countries, many of which reported surprisingly low mortality rates: India (37.6), China (0.37), Nigeria (1.4), and Indonesia (59), compared to higher rates in the United States (364), Germany (218), and Brazil (327). Although underreporting may partially account for these differences, the broader patterns remain informative.

We propose the following model:
*Mortality Rate (deaths per 100,000)~F (Governance, GDP Per Capita, Health, Expenditure Per Capita)*

The dependent variable is national COVID-19 mortality per 100,000 population. Governance performance is operationalized through categorical dummy variables representing different performance tiers. To isolate the effect of governance from broader economic conditions, GDP per capita and health expenditure per capita are included as control variables. These adjustments allow for an evaluation of whether governance performance influences mortality outcomes independently of income and health system investment.

The model is heuristic in nature, designed to explore associations rather than to test a formal theory. In rapidly evolving contexts, such simplified frameworks can offer actionable insights [9].

We apply Ordinary Least Squares (OLSs) regression to estimate the model. Diagnostic checks were conducted prior to analysis: the Breusch–Pagan test indicated no heteroskedasticity, and the Durbin Watson statistic confirmed the absence of autocorrelation. Regression results are presented in Table 5.

The model demonstrates strong explanatory power with an adjusted R^2^ of 0.72. All coefficients are statistically significant at the 1% level and exhibit the expected directional signs. High governance performance is associated with a reduction of approximately 88.6 deaths per 100,000 population relative to the baseline (intercept = 175 deaths/100,000). Surprisingly, low governance performance is associated with 15 fewer deaths per 100,000, which we interpret as likely due to mortality underreporting in several large low-income countries, despite the coefficient being statistically significant.

Control variables showed the expected effects: a 1% increase in income per capita is associated with a 0.24% reduction in deaths per 100,000, and a 1% increase in health expenditure corresponds to a 0.16% decline. These findings highlight the distinct and complementary roles of economic capacity, health system investment, and governance. Importantly, governance performance remains a strong, independent predictor of COVID-19 mortality, even after adjusting for income and expenditure, underscoring its central role in shaping pandemic health outcomes.

#### 3.2.3. Governance Performance and Vaccine Equity

The COVID-19 pandemic revealed stark inequities in vaccine distribution, with high-income countries (HICs) securing early and abundant access, while many low- and middle-income countries (LMICs) faced significant delays and shortages [17]. This section examines whether national income classification, often correlated with governance quality, significantly influenced vaccine coverage rates.

We utilized the IMF-WHO COVID-19 Vaccine Tracker (IMF & WHO, n.d.), which compiles data on vaccine doses secured by each country, acquisition mechanisms (e.g., COVAX or regional initiatives), and key demographic and economic indicators, including population size and income level. Countries are classified following the World Bank system into Low-Income (LIC), Lower-Middle-Income (LMIC), Upper-Middle-Income (UMIC), and High-Income (HIC) categories. Vaccine coverage is operationalized as the number of doses secured per capita.

Given that income level often reflects underlying governance capacity [9], we explore whether income classification functions as a proxy for broader institutional performance in shaping vaccine access and preparedness.

Table 6 presents summary statistics showing that high-income countries secured an average of 482 vaccine doses per 1000 people, compared to just 135 doses in lower-income countries.

Figure 6 highlights stark inequities in global vaccine allocation, with high-income countries securing substantially greater coverage compared to low- and middle-income groups.

We conduct a one-way ANOVA test to determine whether the observed disparities are statistically significant. The outputs in Table 7 show a *p*-value of 0; we find the difference of vaccine coverage across income groups highly statistically significant.

For completeness, we conduct pairwise comparisons using the Tukey Honest Significant Difference (HSD) test. Results, presented in Table 8, show that the differences in vaccine coverage between every pair of income groups are highly statistically significant. These findings provide strong evidence that disparities in vaccine coverage across income levels are both widespread and systematic.

To assess whether vaccine coverage disparities are linked to governance performance independently of income, we estimate the following model:
Vaccine Coverage~F (Governance Performance, Per Capita Income)

Including per capita income as a control variable allows us to isolate the effect of governance performance from that of national wealth. We use the same World Governance Indicators (WGIs) dataset and governance performance levels applied earlier, along with per capita income data from the World Bank. An Ordinary Least Squares (OLS) regression is performed, with diagnostic tests confirming no evidence of heteroskedasticity or autocorrelation. Results, shown in Table 9, indicate that countries in the low governance performance group secured 274 fewer vaccine doses per 1000 population relative to the reference category, which has a baseline of 490 doses (the intercept). In contrast, high governance performance is associated with 323 more doses per 1000. Additionally, a 1% increase in per capita income is associated with a 2.3% increase in vaccine coverage. All coefficients are statistically significant at the 1% level, with signs consistent with theoretical expectations.

### 3.3. Integrated Findings

Table 10 presents the integrated findings from the mixed-methods analysis, synthesizing quantitative results with qualitative insights to explore how governance factors influenced pandemic outcomes. The integration focuses on four thematic areas: vaccine allocation disparities, response timing, funding distribution, and mortality rates. Quantitative data, including regression models, ANOVA, and survival analysis, are paired with qualitative evidence drawn from purposively selected literature. This triangulation enables a more robust understanding of how governance structures, performance levels, and institutional mechanisms shaped both policy decisions and health outcomes during the COVID-19 pandemic.

## 4. Discussion

The COVID-19 pandemic exposed critical weaknesses in public health administration, ranging from delayed interventions in vulnerable populations to the implementation of economically burdensome lockdown strategies. These failures emphasized the role of governance capacity as a determinant of effective crisis response.

This study contributes to growing empirical and theoretical evidence that governance plays a pivotal role in shaping public health outcomes during global crises. By operationalizing governance through the Government Effectiveness and Regulatory Quality dimensions of the World Governance Indicators (WGI), we provide a structured, quantifiable approach to assessing institutional capacity. The strong and statistically significant associations observed between governance quality, mortality rates, and vaccine coverage, even after controlling for income and health sector spending, underscore the centrality of governance in pandemic responsiveness.

Our findings align with key theoretical frameworks. State capacity theory highlights the ability of institutional systems to deliver public goods under stress [44], while institutional theory emphasizes how formal administrative structures and rules influence policy coherence and legitimacy. Countries with higher governance scores in our dataset consistently enacted more timely interventions and secured greater vaccine access, confirming that institutional effectiveness is not merely symbolic; it is a functional component of resilience.

This evidence also reinforces insights from the health systems resilience literature, which underscores the importance of adaptive capacity, transparency, and coordination in emergency response [45]. Governance is often cited as a core ‘building block’ of resilient systems [46], yet it remains undermeasured in empirical research. By quantifying governance performance and linking it directly to measurable outcomes, this study addresses that gap and provides an evidence-based foundation for future reform efforts.

While the primary focus of this study was not on regime type, the role of democratic governance in shaping effective pandemic responses warrants further attention. Democratic systems, characterized by pluralistic decision-making, institutional checks, and public accountability, may facilitate more transparent and participatory health governance, albeit sometimes at the cost of initial response speed [9]. Conversely, more centralized or authoritarian systems can act swiftly but may lack inclusive deliberation or long-term legitimacy [47].

The capacity for political learning, defined as the ability of political institutions to internalize crisis lessons and adapt accordingly, is uneven across governance models. As Kickbusch and Leung [48] argue, institutional memory and political will are essential to embedding resilience into health systems beyond immediate emergency contexts.

This observation aligns with our findings, which suggest that governance effectiveness during the COVID-19 pandemic may be contingent not only on structural capacity but also on the normative orientation of the political system.

### 4.1. Limitations

Limitations include potential underreporting of mortality in some low-income countries, the inability to account for all confounding variables, and the reliance on secondary data.

### 4.2. Recommendations

Integrate governance performance metrics into pandemic preparedness monitoring frameworks and treaty implementation.

National governments should prioritize sustained investment in core governance functions to strengthen public health institutions.

Reform international instruments to include enforceable governance standards that enhance equity, accountability, and the speed of response during health crises.

Promote a policy shift toward institutional reform at both national and global levels to ensure that health systems are not only adequately resourced but also effectively governed.

Future studies should investigate the intersection between democratic governance, policy memory, and sustained health system reform.

## 5. Conclusions

Governance is frequently cited in global health policy discourse, but often without the empirical substantiation needed to inform reform. This study helps bridge that gap by demonstrating statistically robust associations between governance performance and key pandemic outcomes, namely, timely response, mortality rates, and vaccine distribution.

Our findings highlight that improving pandemic preparedness requires more than financial investment; it also depends on institutional frameworks that enable efficient, accountable, and timely responses. This underscores that governance is not simply a contextual backdrop; it is a decisive mechanism shaping health outcomes.

As the global health community continues to develop post-COVID preparedness strategies, reforming governance structures must be treated as a strategic priority. Strengthening institutional capacity at both national and international levels is essential to prevent the disparities and failures observed during the COVID-19 pandemic. Ultimately, embedding governance reform into future global health planning is not optional; it is foundational to ensuring equity, resilience, and effectiveness in health emergency response.

## Figures and Tables

**Figure 1 ijerph-22-01305-f001:**
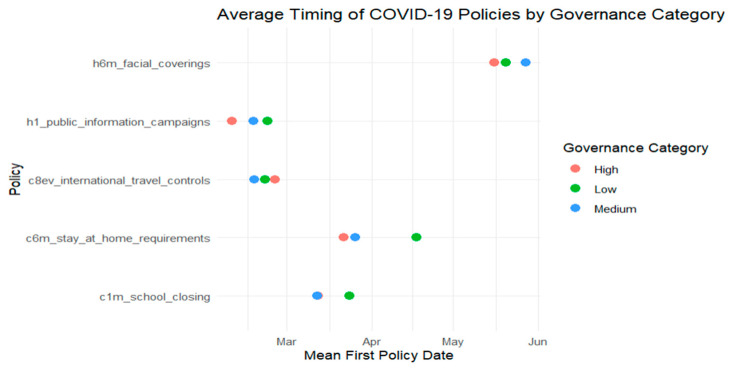
Policy response by governance category (Performance Level).

**Figure 2 ijerph-22-01305-f002:**
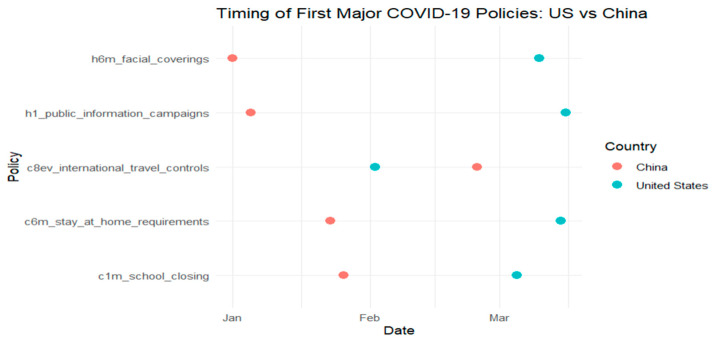
Comparison between the United States and China.

**Figure 3 ijerph-22-01305-f003:**
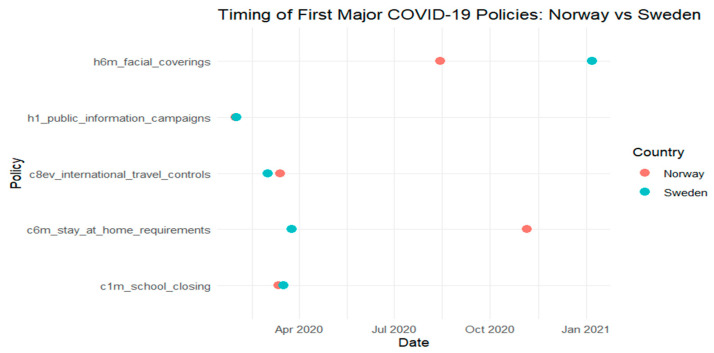
Comparison between Norway and Sweden.

**Figure 4 ijerph-22-01305-f004:**
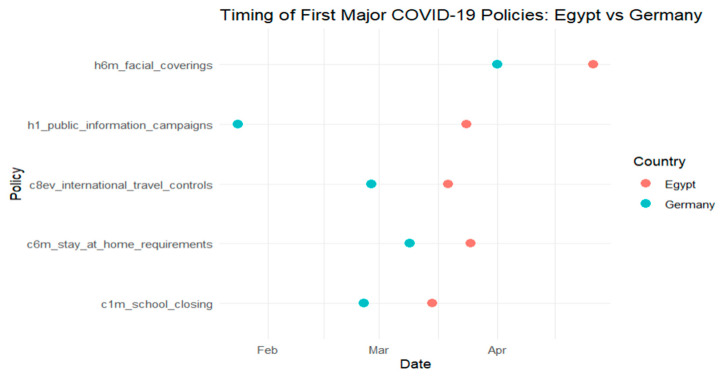
Comparison between Egypt and Germany.

**Figure 5 ijerph-22-01305-f005:**
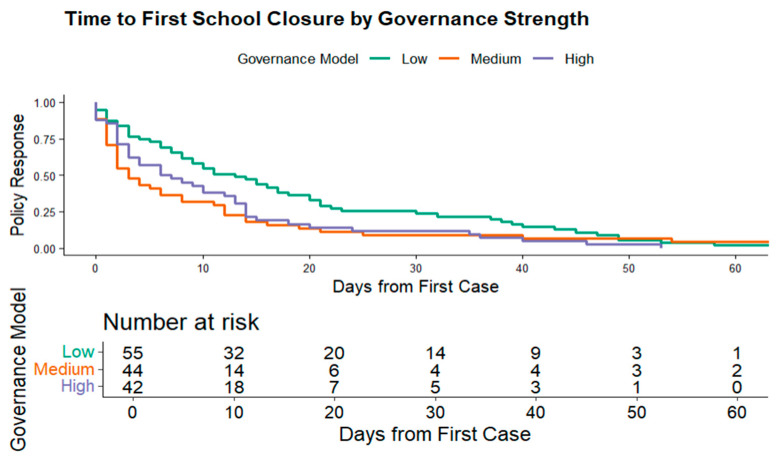
Kaplan–Meier school closure survival curve (3 governance performance levels).

**Figure 6 ijerph-22-01305-f006:**
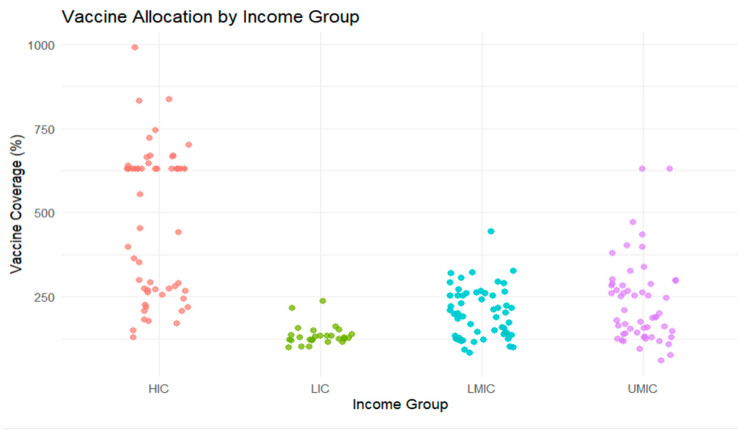
Vaccine allocation by income group.

**Table 1 ijerph-22-01305-t001:** Coding framework.

Themes	Target Research Question	Areas of Analysis	Description
Vaccine Allocation Disparities	Sub-question 1: Governance limitations affecting vaccine equity	Global, regional, and national decision-making processes	Analyze which actors influenced vaccine allocation and the frameworks guiding those decisions.
		Role of COVAX and bilateral agreements	Evaluate the effectiveness of global initiatives and bilateralism in equitable distribution.
		Political and economic access barriers	Investigate how geopolitical factors, funding disparities, and national interests influenced vaccine accessibility.
		Transparency in procurement and allocation	Examine transparency in vaccine procurement and its impact on equitable access.
Pandemic Response Time	Sub-question 1: Governance limitations impacting response time	Effectiveness of global coordination mechanisms (e.g., WHO emergency declarations)	Evaluate how international health entities and national governments coordinated pandemic responses.
		Delays in international pandemic response	Identify structural, political, and logistical barriers that led to delays in response activation.
		Centralized vs. decentralized governance models	Analyze how different governance structures impacted response efficiency.
		Role of legal frameworks (e.g., IHR)	Examine how legal tools helped or hindered rapid responses.
Global Health Funding Distribution	Sub-question 1: Governance limitations impacting funding distribution	Governance mechanisms behind funding allocation	Investigate the decision-making processes guiding pandemic funding.
		Disparities in funding distribution	Assess equity in access to financial resources across countries.
		Role of IMF, World Bank, and national governments	Analyze contributions and limitations of key financial institutions.
		Accountability and transparency	Evaluate mechanisms used to monitor and track financial flows.
Mortality Rates by Governance Model	Sub-question 2: How different governance models influenced pandemic outcomes?	Impact of public health interventions on mortality rates	Assess the effectiveness of public health strategies in reducing deaths.
		Influence of governance models on emergency response	Examine the extent to which governance models supported or hindered healthcare delivery.
		Case studies of successful vs. failed approaches	Compare countries with effective vs. ineffective governance outcomes.

**Table 2 ijerph-22-01305-t002:** Statistical methodology for quantitative analysis.

Section	Variable/Outcome	Statistical Test	Data Sources
1. Governance Performance Level (GPL) and Response Timing	School Closure Timing	Kaplan–Meier survival analysis; log-rank test to compare groups.	Oxford COVID-19 Government Response Tracker (OxCGRT) and World Governance Indicators (WGIs).
2. Governance Performance Level (GPL) and Health Outcomes	COVID-19 Mortality Rate	Ordinary Least Squares (OLSs) regression. Breusch–Pagan (heteroskedasticity), Durbin–Watson (autocorrelation).	Data: WHO COVID-19 Mortality Database, World Bank, and WGI
3. Governance Performance Level (GPL) and Vaccine Equity	Vaccine Coverage Across Income Groups	One-way ANOVA with Tukey HSD post hoc comparisons.	Data: Our World in Data, UNICEF, GAVI, WHO, and COVAX reports.
	Vaccine Coverage across Governance Performance Levels (GPLs)	OLS regression with governance dummies and GDP controls.	Data: WHO, World Bank, and WGI

**Table 3 ijerph-22-01305-t003:** Sample studies.

Vaccine Allocation Disparities	
Wouters et al., 2021 [13]	The study aims to analyze the barriers to global COVID-19 vaccine access by assessing challenges in production, affordability, allocation, and deployment and exploring potential policy solutions.
Eccleston-Turner and Upton (2021) [14]	Evaluates global vaccine-sharing mechanisms
de Bengy Puyvallée and Storeng, 2022 [15]	Examines how political interests shaped vaccine distribution under COVAX, leading to disparities in access.
Chiriboga et al., 2020 [16]	It calls for ethical global leadership to address health inequities.
Harman S. et al., 2021 [18]	Critiques of global vaccine inequities, arguing that donor-based approaches reinforce dependency and advocating for reparative justice through intellectual property waivers and local manufacturing.
Bayati et al., 2022 [19]	Factors affecting inequality in the distribution of the COVID-19 vaccine (macro-level and micro-level factors).
Web et al., 2022 [20]	The study examines the lack of transparency in pharmaceutical pricing, highlighting how confidential agreements, undisclosed profit margins, and information asymmetry disadvantage low- and middle-income countries (LMICs) in vaccine procurement.
Pandemic Response Time	
Elekyabi, S.Y. 2020 [21]	To evaluate the role and response of the World Health Organization (WHO) during the COVID-19 pandemic from the perspective of international law, with a focus on legal authority, accountability, transparency, and compliance mechanisms.
Cortes, Pacheco, and Fronteira, 2024 [22]	To emphasize the role of social, political, and economic factors and argue for the integration of social science insights into pandemic preparedness and response strategies.
Baum, F., et al. 2024 [23]	The primary objective of the study is to investigate how political factors influenced intersectoral actions during the COVID-19 pandemic, focusing on the interplay of ideas, interests, and institutions.
Agyapon-Ntra and McSharry, 2023 [24]	To empirically assess the effectiveness of government non-pharmaceutical interventions (NPIs) on COVID-19 spread and compliance, considering demographic and socioeconomic factors globally.
Navarro, C., and Velasco, F., 2022 [26]	The study argues that Spain’s response to COVID-19 marked a shift from centralized control to innovative multilevel coordination, revealing the adaptability of its decentralized governance. It highlights how crisis management required both recentralization and cooperative intergovernmental mechanisms
Ahmad, E. 2020 [27]	To examine China’s multilevel governance and fiscal framework in responding to COVID-19 and to compare with international responses across governance models.
Sun, Y., et al., 2021 [28]	The study aims to extract lessons from China’s COVID-19 response—especially in government action, transmission pathways, and medical resource allocation—and to advocate for the creation of a global network that enhances resilience to health risks through international coordination and cooperation.
GIGA Focus Asia, 2025 [29]	To examine India’s COVID-19 response from a federal governance perspective, using case studies from four states (Kerala, Odisha, Karnataka, and Uttar Pradesh), and assess how decentralization, leadership, coordination, and community engagement influenced crisis outcomes.
Angelici et al., 2023 [30]	To examine how different multilevel governance strategies—centralized (Spain) versus decentralized (Italy)—affected pandemic outcomes (infections, hospitalizations, ICU admissions, deaths) during the first wave of COVID-19.
*Global Health Funding Distribution*	
Sridhar and Batniji, 2008 [31]	To assess how major global health donors allocated funds and whether disbursements are aligned with global disease burden. It advocates for transparency and improved priority setting in global health governance
Fergus, 2020 [32]	To explore the structure of global health aid by analyzing financial flows and relationships between funders, intermediaries, and recipients over 1990–2017, identifying key gaps and governance challenges
Goosby E., 2019 [33]	The article explores the governance mechanisms of the Global Fund, emphasizing its effectiveness not only in fighting diseases but also in contributing to enhanced governance at the country level.
Kavanagh and Chen, 2019 [34]	To assess how the Global Fund’s governance model has influenced civil society participation, political accountability, and broader governance practices in recipient countries
Tichenor, 2021 [35]	To critically examine how the World Bank shapes global health through its roles in knowledge production, governance, and financing, and to develop a research agenda that promotes greater accountability and responsiveness to local health priorities.
Ruger, 2005 [36]	To trace the evolution of the World Bank’s involvement in global health from its original post-WWII reconstruction mission to its present role as the largest external health funder and to analyze the theoretical, political, and institutional shifts that shaped this transition.
Moosa, N., 2018 [37]	To assess whether IMF loan conditionalities have a positive or negative impact on government health expenditure and to argue that these conditionalities tend to reduce such expenditure, despite IMF claims.
Sobhani, S., 2019 [38]	To evaluate the historical and current impacts of IMF and World Bank policies on health outcomes in developing countries, especially focusing on privatization and health system strengthening (HSS) initiatives
Chuan De Foo, et al. 2023 [39]	To provide an overview of health financing policies adopted in 15 countries during the COVID-19 pandemic, develop a framework for resilient health financing, and use the pandemic as a case to advocate for progress towards universal health coverage (UHC).
Mortality Rates by Governance Model	
Fukuyama, F., 2020 [9]	To argue that the quality of governance, rather than regime type (democracy vs. autocracy), is the key determinant of a country’s effectiveness in responding to the COVID-19 pandemic.
Lim et al., 2023 [40]	To examine the reasons behind varying COVID-19 mortality outcomes among countries that implemented similar containment policies, with a focus on how responsiveness and national culture influenced these differences

**Table 4 ijerph-22-01305-t004:** ANOVA—analysis of variance.

Source	Df	Sum of Squares (SS)	Mean Square (MS)	F Value	*p*-Value
Governance Performance Level	1	27,328.314	12,061.35	5.85	0.01
Residuals	139	35,351.306	4401.86		

**Table 5 ijerph-22-01305-t005:** Mortality Regression Model.

	Adjusted R^2^	Reference Category		
Mortality/100,000	0.72	Medium Governance Performance Level		
Independent Variables	Estimate	Std. Error	t-value	*p*-value
Intercept	175.01	35.64	4.91	<0.001
Low Governance Performance Level	−15.21	6.20	−2.45	0.018
High Governance Performance Level	−88.55	29.39	−3.01	0.004
Income per Capita	−0.24	0.06	−4.01	<0.001
Health Expenditure per Capita	−0.16	0.07	−2.28	0.025

**Table 6 ijerph-22-01305-t006:** Vaccine coverage summary statistics.

Income Group	Count	Mean	Median	Standard Deviation
LIC	27	135	128	31
LMIC	55	202	201	75
UMIC	55	230	188	123
HIC	59	482	629	217

**Table 7 ijerph-22-01305-t007:** Vaccine coverage & income (ANOVA).

	Degrees Freedom	Sum Squares	Mean Squares	F Value	*p* Value
Income Group (Between Groups)	3	3,443,472	1,147,824	56.87	0
Residuals (Within Groups)	192	3,868,283	20,147		

**Table 8 ijerph-22-01305-t008:** Vaccine Coverage Tukey test.

Classification	Difference	Lower	Upper	*p*-Adjusted
LIC-HIC	−347.11	−432.59	−261.64	0.00
LMIC-HIC	−280.11	−349.06	−211.16	0.00
UMIC-HIC	−251.84	−320.79	−182.89	0.00
LMIC-LIC	67.01	−19.44	153.45	0.02
UMIC-LIC	95.27	8.83	181.71	0.02
UMIC-LMIC	28.26	−41.89	98.41	0.07

**Table 9 ijerph-22-01305-t009:** Vaccine governance regression.

Dependent Variable	Adj R-Square		
Vaccine Coverage	0.68		
Independent Variables	Coefficient Estimate	Standard Error	T-Statistic
Intercept	489.9	47.13	10.39
Governance Low	−274.5	41.61	−6.60
Governance High	323.38	46.09	7.02
Income Per Capita	2.3	0.60	3.83

**Table 10 ijerph-22-01305-t010:** Integrated findings from the mixed-methods analysis.

Quantitative Findings	Qualitative Insights	Integrated Interpretation
1-Vaccine Allocation Disparities		
The analysis of vaccine doses secured per 1000 population across income groups revealed stark disparities. ANOVA and Tukey post hoc tests confirmed statistically significant differences (*p* < 0.001), with high-income countries securing a mean of 482 doses compared to 135 for low-income countries. OLS regression demonstrated that governance performance level, independent of income, significantly influenced vaccine coverage. A low-governance classification was associated with 274 fewer doses per 1000 (*p* < 0.01).	The purposive review highlighted political and economic barriers to equitable vaccine access. Studies emphasized how high-income countries engaged in vaccine nationalism and bilateral deals, bypassing multilateral initiatives like COVAX [15,16]. Transparency deficits in procurement and opaque pharmaceutical pricing disadvantaged LMICs [31]. Governance gaps in global coordination were a recurring theme.	The quantitative disparities are supported by qualitative evidence of systemic governance failures, including fragmented procurement frameworks and geopolitical self-interest. This triangulation confirms that governance capacity, regardless of income, influenced vaccine equity.
*2*-Pandemic Response Time		
Kaplan Meier survival analysis showed that high-governance performance level countries implemented school closures and other interventions significantly earlier than low-governance countries (log-rank *p* = 0.007). ANOVA tests also showed a statistically significant difference in intervention timing across governance categories (*p* = 0.01).	Elekyabi [21] and Ahmad [27] stressed the WHO’s limited authority to enforce early action and the importance of centralized governance models in enabling swift response. Comparative case studies from China, Spain, and Italy illustrated how coordination mechanisms and institutional trust facilitated or hindered timely interventions.	The survival analysis quantitatively confirms the narrative that governance quality dictates response speed. Centralized, high-capacity systems responded faster, validating the qualitative critique of decentralized inefficiencies and political fragmentation.
3-Global Health Funding Distribution		
Quantitative analysis was excluded due to statistically insignificant findings. This may reflect the nature of the data, which primarily captured donor-based funding, often allocated independently of governance performance.	Studies identified conditionalities from institutions like the IMF as barriers to flexible funding [37,38]. Governance structures within the Global Fund and Gavi promoted inclusivity but often faced power asymmetries. Lack of transparency in donor disbursements remained a persistent issue [37]	Qualitative evidence suggests that weak institutional governance created administrative bottlenecks, limiting timely access to available pandemic-related funding.
4-Mortality Rates by Governance Performance Level		
OLS regression showed that governance performance level had a strong and significant effect on COVID-19 mortality. High-governance countries had 88.6 fewer deaths per 100,000 than low-governance ones, with an R-squared of 0.72, indicating high model fit.	Lim [40] and Fukuyama [9] emphasized that responsive governance, institutional trust, and public engagement were crucial in mitigating mortality. Countries like South Korea and New Zealand demonstrated that timely, coordinated public health interventions reduced death rates despite differing income levels.	The quantitative evidence validates qualitative case studies showing that legitimacy, trust, and agility in governance significantly shaped mortality outcomes.

## Data Availability

Available upon request.

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
