# Peer review of "Governance in Crisis: A Mixed-Methods Analysis of Global Health Governance During COVID-19"

_ijerph, 2025, doi:10.3390/ijerph22081305_

Round 1
Reviewer 1 Report
Comments and Suggestions for Authors
My general assessment of this study is as follows, first of all, it is noticeable that this study does not take into account the journal's spelling rules in terms of form. First of all, researchers should comply with these spelling rules, also:
1- 1.2 Research Problem is kept quite short, it should be expanded.
2- 1. Introduction should be more detailed, the expressions should be made more understandable.
3- 2.1.1 Study Design should be detailed
4- Table 1: Coding Framework should be arranged in a proper table.
5- Table 2: Statistical Methodology for Quantitative Analysis should be shown on a single page.
6- 3.1 Qualitative title is left blank, what will be done in the subheadings should be stated with a sentence.
7- In general, the subheadings are subject to an irregular alignment and this is the case throughout the study. For example, 3.1.3.2. and 3.1.3.3.
8- Table 4: Sample Studies table title should be added to the page where the table is located.
9- There are differences between the page intervals between the columns, such as between lines 343-354 and lines 355-361.
10- Figure 1: Policy Response By Governance Category (Performance Level) cannot be understood, the figure should be made understandable within the framework of the writing rules.
11- Figure 2: Comparison between United States and China cannot be understood, the figure should be made understandable within the framework of the writing rules. This situation is also valid for other figures.
12- In general, tables should be standardized, the shape of each table is quite different from the other.
13- Only the table is given in the title 3.3. Integrated Findings, and what this table means should be explained before moving on to the table.
Reviewer 2 Report
Comments and Suggestions for Authors
I have thoroughly reviewed the manuscript investigating the effectiveness and limitations of global health governance during COVID-19. While the topic seems interesting, the manuscript conducted through a big missing that is not considered, so the result does not present a novel or sufficiently distinct perspective. I expected a more original contribution to this area of research.
I have several concerns regarding the manuscript’s structure, methodology, and overall presentation, which currently do not meet the standards for publication in the IJERPH.
Although the manuscript focuses on a mixed method, it fails to present and elaborate essential contextual factors. Global health governance varies widely — from state-level to country-level and international, each with differing scales, responsibilities, authorities, facilities, and target populations. Lacking the consideration of this omission leads to significant limitations in the research methodology, particularly in the selection of study areas, data collection instruments, sample representation, and analytical approach.
This study tried to focus on limitations in global health governance affecting decision-making, mortality rates, and equitable vaccine distribution during the pandemic, and explored governance effectiveness, independent of income classification, that influences pandemic response outcomes across centralized and decentralized systems, but the foundational information is missing.
The review of literature is very poor, and there is no classification or details about the Oxford COVID-19 Government Response Tracker (OxCGRT), the World Bank’s Worldwide Governance Indicators (WGI), and health spending records during COVID-19.
While the OxCGRT report indicates 24 policy indicators and a miscellaneous notes field of government response organized into four groups (C - containment and closure policies, E - economic policies, H - health system policies, and V - vaccination policies), the study never reveals the study scope and framework. Furthermore, there is no elaboration on how the World Bank’s Worldwide Governance Indicators (WGI) and health spending records are integrated in this study. Thus, there is no proof to confirm that the integration of OXCGRT and WGI, and health spending records is valid and reliable.
The abstract also omits the research contribution and fails to identify the knowledge gap addressed by the study. The introduction does not articulate the specific or unique challenges investigated, nor does it clarify what distinguishes this work from prior research. We know that the OxCGRT was conducted over the world for two years, but what has been added by this study?
The problem statement is a futile text that the readers will not miss, nothing is we remove it from manuscript.
Research methodology is one of the most problematic aspects of the manuscript. The rationale behind the selection of the mixed method is not explained, and the type of mixed method is unclear.
Another issue refers to inclusion criteria. The manuscript included all published papers from peer review journals, but there are many journals that are not really peer review and looking for APC. Thus. The inputs are not valid. The authors must include the papers indexed in reputable database (Scopus, WoS, Pub-Med) There is no information and details about keywords to explore the papers. How were the papers searched?
In the quantitative section, the manuscript selected three following phases from different sources including:
Governance Performance Level (GPL) and Response Timing
Governance Performance Level (GPL) and Health Outcomes
Governance Performance level (GPL) and Vaccine Equity
But there is missing to elaborate the data harmonization, standardization, cross-source validation for employing in this study.
The results are unclear and there is no clear connection between the results of review (published papers) and secondary data analysis.
A clear description about the findings of qualitative part is needed to highlight the impact of geographical and economy status of regions on response timing, health outcomes, and vaccine equity.
Finally, the conclusion section needs significant improvement. It lacks a clear synthesis of the findings, both empirically and theoretically. The current version does not effectively generalize the results or discuss their implications in a broader context.
In summary, this manuscript fails to offer a significant advancement to existing literature due to its weak theoretical foundation, methodological shortcomings, and lack of clarity in presentation. I recommend substantial revisions before they can be reconsidered for publication.
Author Response
Please see the attachement

Reviewer 3 Report
Comments and Suggestions for Authors
I find the study very valuable because it attempts to examine the quality of governance in the management of the global pandemic. Its methodology is not mature enough, although epidemiological and, within this, mortality rates are undoubtedly related to the ability of a government to analyze evidence and make decisions based on it, to global cooperation, to prepare the healthcare system for crises "in peacetime", and whether health is a priority in government policy at all. In addition, it is also a consideration how democratic the nature of the exercise of power is in a given country. The authors tried to create a picture of the role of governance at different levels with the help of multi-directional quantitative and qualitative techniques, carry out a so-called purposive review, and took into account the response and support of international organizations. Overall, the draft correctly states that national health governance was not effective enough in most places, and professional aspects were pushed into the background by political and economic considerations. The big question, which can perhaps be implicitly read from the extensive working material, is whether the world and nation states are learning from the pandemic, and to what extent are politicians forgetting.
Round 2
Reviewer 1 Report
Comments and Suggestions for Authors
My general assessment of the study is as follows:
1- The text in numbers 12-261 is not justified (pages 1 to 7).
2- Similarly, numbers 415-416 are not justified (page 10).
3- Similarly, numbers 486-507 are not justified (page 14).
4- Similarly, the text in pages 14 to 25 is not justified.
In other words, the text in the study should generally be justified.
Furthermore, the tables and figures should be aligned.
Reviewer 2 Report
Comments and Suggestions for Authors
Dear Authors,
I have gone through the revised version of the manuscript and the authors’ responses. It is evident that the authors made a sincere effort to improve the manuscript based on the feedback, and I acknowledge the improvements made in both content and clarity.
Most of the comments have been addressed, and I believe the manuscript meets the minimum requirements for publication in the journal and can be recommended for acceptance.
Author Response
Dear Reviewer,
Thank you very much for your positive assessment and for acknowledging the revisions made. We sincerely appreciate your thoughtful feedback throughout the review process, which has greatly contributed to strengthening the quality, clarity, and rigor of the manuscript. I am pleased to hear that the revised version meets the journal’s standards and is now considered suitable for publication.